# A Search for Tick-Associated, Bronnoya-like Virus Spillover into Sheep

**DOI:** 10.3390/microorganisms11010209

**Published:** 2023-01-13

**Authors:** Bianca Elena Bratuleanu, Cristian Raileanu, Delphine Chrétien, Pablo Guardado-Calvo, Thomas Bigot, Gheorghe Savuta, Sarah Temmam, Marc Eloit

**Affiliations:** 1Pathogen Discovery Laboratory, Institut Pasteur, 75015 Paris, France; 2Regional Center of Advanced Research for Emerging Diseases, Zoonoses and Food Safety (ROVETEMERG), “Ion Ionescu de la Brad”, Iasi University of Life Sciences, 700490 Iași, Romania; 3WOAH (OIE) Collaborating Centre for Detection and Identification in Humans of Emerging Animal Pathogens, Institut Pasteur, 75015 Paris, France; 4Department of Virology, Structural Virology Unit, CNRS UMR 3569, Institut Pasteur, Université Paris Cité, 75015 Paris, France; 5Bioinformatics and Biostatistics Hub, Institut Pasteur, Université Paris Cité, 75015 Paris, France; 6Department of Biological and Pharmaceutical Sciences, Alfort National Veterinary School, 94700 Maisons-Alfort, France

**Keywords:** sheep, ticks, spillover, metatranscriptomics, LIPS

## Abstract

Tick-borne diseases are responsible for many vector-borne diseases within Europe. Recently, novel viruses belonging to a new viral family of the order *Bunyavirales* were discovered in numerous tick species. In this study, we used metatranscriptomics to detect the virome, including novel viruses, associated with *Ixodes ricinus* collected from Romania and France. A bunyavirus-like virus related to the Bronnoya virus was identified for the first time in these regions. It presents a high level of amino-acid conservation with Bronnoya-related viruses identified in *I. ricinus* ticks from Norway and Croatia and with the *Ixodes scapularis* bunyavirus isolated from a tick cell line in Japan in 2014. Phylogenetic analyses revealed that the Bronnoya viruses’ sub-clade is distinct from several *Bunyavirales* families, suggesting that it could constitute a novel family within the order. To determine if Bronnoya viruses could constitute novel tick-borne arboviruses, a Luciferase immunoprecipitation assay for detecting antibodies in the viral glycoprotein of the Romanian Bronnoya virus was used to screen sera from small ruminants exposed to tick bites. No positive serum was detected, suggesting that this virus is probably not able to infect small ruminants. This study represents the first serological investigation of mammalian infections with a Bronnoya-like virus and an initial step in the identification of potential new emergences of tick-borne arboviruses.

## 1. Introduction

Ticks are widespread across Europe and are the primary arthropod vector of both human and domestic animal disease agents [1]. In terms of public health, the most important European tick is *Ixodes ricinus*, which can transmit many pathogens, including bacteria, parasites and viruses, due to specific biological adaptations and its capacity to feed on numerous different animal species [2,3]. 

High-throughput sequencing (HTS) has become more frequently used, allowing for a better understanding of the infectious agents carried by ticks. The virome of ixodid ticks is actively being studied, and recent research has indeed reported the identification of new tick-borne viruses, some of which are pathogenic, such as the Heartland virus and the Severe Fever with Thrombocytopenia Syndrome virus. Viruses representing unknown viral families have also been reported, including those for which the risk of zoonotic transmission to humans or domestic animals has not yet been determined [4,5].

Bunyaviruses constitute an extremely diverse group of RNA viruses infecting arthropods, protozoans, plants, and animals. They have been recently re-categorized by the International Committee on Taxonomy of Viruses (ICTV) to the order *Bunyavirales,* which comprises thirteen different viral families [6]. Of these, three of families include human pathogens transmitted by arthropod vectors (*Peribunyaviridae, Nairoviridae,* and *Phenuiviridae*) and one transmitted through contact with rodents’ urine and feces (*Hantaviridae*). Due to the increased frequency of outbreaks and the spread of competent vectors for bunyaviruses, the World Health Organization has designated several bunyaviruses as priority pathogens. For example, one of the most severe tick-borne bunyaviruses is the Crimean–Congo Hemorrhagic Fever virus (CCHFV), a nairovirus responsible for viral hemorrhagic fever in humans and abortions in livestock. The main vector of CCHFV is the *Hyalomma* tick species. The virus affects both humans and livestock in various regions of the world including Africa, Asia, Eastern and South-eastern Europe and, more recently, Southwestern Europe [7]. Another highly pathogenic bunyavirus is the Severe Fever with Thrombocytopenia Syndrome virus (SFTSV) [8], a phlebovirus transmitted by *Haemaphysalis longicornis* ticks which is responsible for hemorrhagic fever in humans. The virus was first identified in humans in 2007 in the Chinese province of Henan, and since then its distribution has expanded, being reported in East Asia and North America. It was recently named Dabie bandavirus by the ICTV. Its case–fatality rate ranges from 2.7% to 45.7%. Heartland virus (HRTV), a phlebovirus phylogenetically close to SFTSV, was reported in 2009 in North America with a clinical presentation similar to SFTSV. The current distribution of HRTV disease cases closely mirrors the distribution of the lone star tick, *Amblyomma americanum* [9].

Novel *Bunyavirales*-related sequences are continuously reported due to an increased knowledge of the viral communities infecting ticks. For example, sequences belonging to a new *Bunyavirales*-like virus that seems to lack the S-segment were recently identified in *Ixodes* sp. ticks collected from different countries and biotopes. The Bronnoya virus was identified for the first time in 2017 in *I. ricinus* ticks from Norway [4] and in Croatia in 2022 [10]. Related—but more distant—viruses were reported in *Dermacentor* and *Hyalomma* sp. ticks collected in 2019 from Chinese cattle and camels [11] (Fuhai tick bunyavirus) and in *Hyalomma marginatum* ticks collected in 2018 from Russia (Volzhskoe tick virus). In addition, a virus close to the Bronnoya virus was identified and observed through electron microscopy in the *Ixodes scapularis* embryo-derived cell line ISE6 [12]. Therefore, it is questionable if Bronnoya-related viruses can infect vertebrates or other tick species, considering that these viruses have only been identified in *Ixodes* sp. ticks thus far.

This work summarizes the data from long-term studies carried out in France (in the Alsace and Ardennes regions) and Romania (in the Danube Delta and Iasi regions) between 2010 and 2021 on viral communities carried by *I. ricinus* ticks. It includes the discovery and comprehensive characterization of a new tick-borne virus related to *Bunyavirales* with an unknown pathogenic potential for which a serological tool was implemented to evaluate the exposition of vertebrate animals to this virus.

## 2. Materials and Methods

### 2.1. Tick Collection and Preparation of Metatranscriptomics Libraries

#### 2.1.1. Ticks Collected in France, 2010–2012

A total of 2236 questing adult and nymph ticks were collected via flagging from northeastern France between 2010 (Alsace) and 2012 (Ardennes) and processed as previously described [5]. Briefly, ticks were washed to remove external contaminants. The nymphs were pooled into groups of 15 individuals (116 pools in total) and adults were treated individually. Tick materials were crushed into Dulbecco’s modified Eagle’s medium, supplemented with 10% fetal calf serum, before a total RNA extraction with the Nucleospin RNA II kit (Macherey-Nagel, Duren, Germany), performed according to the recommendations of the manufacturer.

The RNA samples were pooled according to their geographical location and reverse-transcribed into cDNA using the SuperScript III Reverse Transcriptase and random hexamers (Invitrogen, Carlsbad, CA, USA). The tick material was then randomly amplified using the multiple displacement amplification (MDA) protocol with phi29 polymerase and random hexamers as described previously [13]. Library preparations and sequencing with an Illumina HiSeq2000 (Illumina Inc., Saint Diego, CA, USA) sequencer on a 2 × 100 bp format were outsourced to DNAVision Company (Charleroi, Belgium).

#### 2.1.2. Ticks Collected in Romania, Iasi County, 2015

A total of 315 ticks were collected by flagging from March to September of 2015. Ticks were collected from six collection sites in Bucium, C.A. Rosetti, Breazu, Ciric, Cetăţuia and Bârnova, representing suburban sites intended for recreational activities in Iasi County, Romania. These are regions where human–animal interactions are common.

Ticks were washed to remove any external contaminant, and total RNA was extracted from individual ticks using the Nucleospin RNA II kit (Macherey-Nagel, Duren, Germany) according to the recommendations of the manufacturer. Before sequencing, tick RNA was pooled to form three pools that were processed by the REPLI-g^®^ WTA kit (Qiagen, Hilden, Germany) based on the manufacturer’s specifications. The libraries were sequenced using a paired-ends 2 × 150 bp format and were outsourced to Integragen Company (Evry, France).

#### 2.1.3. Ticks Collected in Romania, Danube Delta Reserve, 2020–2021

A total of 202 ticks were collected in Slava Rusa village (Danube Delta region, Romania) from the body surfaces of adult sheep in October 2020 and May 2021. Ticks were collected at various levels of engorgement on their hosts, including 56 questing (unfed) individuals found questing in the field. Ticks were grouped before extraction, resulting in a total of 20 pools (from three to fifteen ticks/pool). Total RNA was extracted from crushed specimens using the Maxwell^®^ RSC simplyRNA Tissue Kit or the TRIzol Reagent (Invitrogen), followed by the RNeasy mini kit (Qiagen), according to the recommendations of the manufacturers. The 20 pools of tick extracts were then combined to form five NGS libraries using the SMARTer Stranded Total RNA-seq kit v3-Pico input mammalian (Clontech, TaKaRa Bio, San Jose, CA, USA) or the NEBNext Ultra II DNA Library Prep Kit (New England Biolabs, Evry, France). An Agilent Bioanalyzer was used to validate the NGS libraries and the quantification was performed using a Qubit 2.0 Fluorometer (Invitrogen, Carlsbad, CA, USA). Libraries were sequenced using an Illumina Nextseq2000 Sequencer (Illumina Inc., Saint Diego, CA, USA) in a 2 × 100 bp format or in a single-read 1 × 150 bp format onto an Illumina NextSeq500 Sequencer.

### 2.2. Determination of Tick Species

Ticks were first morphologically identified under a stereomicroscope using standard morphological keys [14]. To confirm the identification, we took advantage of the concomitant sequencing of tick transcriptomes and used the Barcode of Life Data Systems (BOLD) as previously described [15]. Briefly, all trimmed reads were mapped onto the *Ixodidae* Cytochrome Oxydase sub-unit I (COI) sequences included in the BOLD database. Mapped reads were de novo assembled, and the resulting contigs were uploaded to the BOLD Identification System (https://www.boldsystems.org/index.php/IDS_OpenIdEngine) (accessed on 30 September 2021) for species identification. Results were finally confirmed by BlastN, (https://blast.ncbi.nlm.nih.gov/Blast.cgi, accesed on 30 September 2021). The ticks investigated in this study were all identified as *Ixodes ricinus*.

### 2.3. Virus Assignation

Raw reads were processed as previously described [16] with an in-house bioinformatics pipeline, Microseek (https://research.pasteur.fr/wp-content/uploads/2022/09/research_pasteur-microseek-a-metagenomic-pipeline-for-virus-diagnostic-and-discovery-microseek-4.mp4; accessed on 20 August 2021) [17], that comprises a quality check and the trimming of raw reads followed by read normalization prior to assembly into contigs and the generation of non-assembled singletons. ORF predictions of both contigs and singletons were performed, and a taxonomic assignation of protein sequences was performed by Blast-based querying using three successive, specialized RVDBs [18] followed by generalist (NCBI/nr and NCBI/nt) databases.

### 2.4. Phylogenetic Analyses

Phylogenetic reconstructions were conducted on the following genes: the conserved, non-structural, RNA-dependent RNA polymerase (RdRP) and the glycoprotein (GP). A MAFFT (Multiple Alignment using Fast Fourier Transform) aligner was used to align the complete ORFs with other viral orders/families’ representative sequences under the L-INS-i (RdRP) or G-INS-I (GP) parameters [19]. Using ATGC Smart Model Selection [20] implemented in http://www.atgc-montpellier.fr/phyml-sms/ (accessed on 3 October 2022) the best amino acid-substitution models that matched the data were selected by utilizing the modified Akaike information criteria. These were LG+G for amino-acid *Bunyavirales* phylogenies and GTR+G for nucleotide phylogenies (whatever segment considered).

Based on the selected substitution model, phylogenetic trees were generated using the maximum likelihood (ML) method provided through the IQ-TREE program (http://iqtree.cibiv.univie.ac.at/; accessed on 3 October 2022) [21]. The nodal support was assessed using the approximate Bayes parameter.

To generate identity matrices between Bronnoya virus genomes, complete RdRP and GP nucleotide and amino-acid sequences were aligned with MAFFT with the same parameters as those used for phylogenetic reconstructions, and matrices were generated using CLC Main Workbench v.21.0.4 (Qiagen, Hilden, Germany; accessed on 5 October 2022).

Molecular evolutionary distances between *Bunyavirales* families were calculated using MEGA7 (https://www.megasoftware.net/; accessed on 5 October 2022) [22]. Briefly, all complete RdRP amino-acid sequences included in the GenBank/RefSeq database were downloaded (MeSH terms “txid1980410 [Organism:exp] AND polymerase NOT partial”) to a total of 353 sequences divided as follows: *Arenaviridae* N = 49; *Cruliviridae* N = 1; *Discoviridae* N = 5; *Fimoviridae* N = 10; *Hantaviridae* N = 36; *Leishbuviridae* N = 1; *Mypoviridae* N = 1; *Nairoviridae* N = 21; *Peribunyaviridae* N = 86; *Phasmaviridae* N = 13; *Phenuiviridae* N = 98; *Tospoviridae* N = 24; *Wupedeviridae* N = 1; and unclassified *Bunyavirales* N = 7. Sequences were aligned with MAFFT with the L-INS-I parameter and grouped according to their recognized family. Distances were calculated between families using the net distance calculation and the *p*-distance algorithm (i.e., MEGA7 took into account the mean distance within a given group).

### 2.5. Sheep and Goats Sera Collection

Between 2019 and 2021, a total of 342 blood samples were collected from small ruminants from six different villages in a rural area within Tulcea County (Danube Delta region) in the same location as the of ticks (Figure 1). Details regarding the number of serum samples and the collection sites are presented in Table 1. These sera were obtained with the approval of the animal owners, under the control of the local veterinary services from Tulcea County. For additional processing, all samples were shipped to the Institut Pasteur in Paris, France.

### 2.6. Serological Screening of Small Ruminants

#### 2.6.1. Antigen Design

To increase the probability of detecting specific antibodies to the Bronnoya-like virus (BroBV-like) and to minimize cross-reactions, we decided to target the viral external protein, meaning the ectodomain of the glycoprotein (GP). As the GP structure of Bronnoya virus is not available in any database, we used AlphaFold to predict the structure of the complete GP chain of both Bronnoya-like viruses identified in Tulcea County, Romania (Figure 2). The program produced a per-residue confidence metric, termed pLDTT, on a scale from 0 to 100. Values of pLDTT higher than 70 reflect reliable models with correct backbone predictions; values below that threshold indicate that the structural prediction is not reliable. We identified the Gc domain (residues 711–1486, average pLDTT = 78) and two sub-domains of Gn: an N-terminal region (residues 1-569) and a C-terminal region (residues 570-664, pLDTT = 85). As in other bunyavirus families, the latter turned out to be the most immunogenic, so we decided to target the C-terminal part of BroBV-like Gn (designed below as the “Gn head”) for serological screenings.

Following this prediction, and because the percentages of amino-acid identity between the Gn heads of the two Bronnoya-like virus strains identified in Tulcea in 2020 differed for more than 15%, it was decided to develop two LIPS assays according to these two strains of Bronnoya-like viruses. Synthetic genes coding for the two heads of Gn were subsequently ordered from GenScript company (Rijswijk, The Netherlands) with codon usage optimized for protein expression in mammalian cells and cloned in the pcDNA3.1(+) vector. An exogenous signal peptide was added to ensure efficient protein secretion. The nanoluc was added to the carboxy-terminal end of the construct, spaced by a 3-residues GSG linker.

#### 2.6.2. Expression of Recombinant Proteins

HEK-293F cells were grown in suspension and transfected with ExpiFectamine 293 reagent (ThermoFisher, Waltham, MA, USA) according to the manufacturer’s recommendations. Briefly, a total of 75.10^6^ cells were harvested and the pellet was resuspended in 25 mL of Expi293 Expression medium before incubation at 37 °C for 30 min. Twenty-five micrograms of each plasmid were diluted in 1.5 mL of Opti-MEM medium and mixed with ExpiFectamine reagent before incubation at room temperature for 20 min. Finally, the mixed ExpiFectamine/plasmid was added to the cells and further incubated at 37 °C under agitation for 4 days. ExpiFectamine Transfection enhancers were added on day 1 post-transfection. Recombinant proteins were harvested directly from the culture supernatant at day 4 post-transfection. Luciferase activity was quantified onto a Centro XS3 LB 960 luminometer (Berthold Technologies, Thoiry, France) by adding 100 µL of NanoGlo reagent (Promega, Madison, WI, USA) to tenfold dilutions of supernatant.

#### 2.6.3. LIPS Assay

A LIPS assay was performed as previously described [23], except that 10^8^ LU of each antigen and 10 µL of non-diluted sera were engaged per reaction. Fifty sera from healthy sheep (kindly provided by Stephan Zientara and Emmanuel Breard, Anses, Maisons-Alfort, France), collected in France in a location distant from where the French ticks were collected, were included in the study as a non-exposed group control. Residual background was defined as the mean of results from ten negative controls (without sera). The positivity threshold was defined as the mean of these controls plus three standard deviations.

### 2.7. Statistical Tests

Statistical analyses were conducted using GraphPad Prism v.8 (GraphPad Software, San Diego, CA, USA). The signal-to-noise light unit (LU) ratios between each group of sera were compared using the Kruskal–Wallis ANOVA and Dunn’s multiple comparisons tests with the French cohort used as a reference.

## 3. Results

### 3.1. Identification and Evolution History of Bronnoya Viruses

In this study, we performed a viral metatranscriptomics analysis of *Ixodes ricinus* ticks collected between 2010 and 2021. The ticks were collected in the environment and during feeding on domestic animals from two countries (France and Romania) with distinct biotopes that range from peri-urban, rural, or forest regions. Among the huge diversity of viruses carried by ticks we have previously described, such as the novel *Coltivirus* [5] or Jingmenviruses [24] identified in French ticks, we detected several strains of a Bronnoya-like virus, a negative-sense, bi-segmented, single-stranded RNA virus which seems to lack the S-segment encoding the nucleoprotein. Bronnoya virus (BroBV) was first reported in 2017, found in *I. ricinus* ticks from Norway that were collected in 2014 [4], and later reported in Croatia in 2022 in *I. ricinus* ticks that were collected in 2012 [10]. Two more distant species of BroBV, still associated with *I. ricinus* ticks, were reported in Russia in 2018 (Almazovo tick virus) and in Croatia in 2012 (*Ixodes ricinus* bunyavirus-like virus 1), [10] along with the the identification of a novel BroBV-related virus in an *Ixodes scapularis* cell line. Here, we identified six novel BroBV-like strains in *Ixodes ricinus* ticks from Alsace and Ardennes (France) and from Iasi and Tulcea counties (Romania) (Table 2).

These strains present a high degree of amino-acid and nucleotide conservation in the RNA-dependent RNA polymerase with other representatives of Bronnoya viruses, ranging from 97.4% to 99.7% at the amino-acid level, and from 91% to 97.4% at the nucleotide level; this is except for Ixodes ricinus bunyavirus-like virus 1, which exhibited a more distant profile than other Bronnoya viruses, with 55% and 60% of amino-acid and nucleotide identity, respectively (Figure 3). As expected, viral glycoproteins presented a lower degree of conservation between BroBV strains when compared to the polymerase, but the same pattern of conservation was observed between Bronnoya viruses (amino-acid identities ranging from 84.4% to 95.9% and nucleotide identities ranging from 76.6% to 91.9%), with the exception of the Ixodes ricinus bunyavirus-like virus 1 that exhibited lower levels of identity (Figure 3). A BroBV-related virus identified in a tick cell line presented a high degree of homology, either at the nucleotide or the amino-acid level, with Bronnoya viruses identified in ticks harvested in Western or Eastern Europe.

Interestingly, phylogenetic reconstruction of the evolutionary history of BroBV L and M segments revealed that the different strains of Bronnoya virus circulating in the same area fall into two sub-clades, as was observed for Romania/Tulcea tick1 and tick2 isolates, suggesting that different viruses are co-circulating in the same tick populations. Of note, the different isolates of BroBV did not cluster similarly according to the segment, further demonstrating that reassortments between Bronnoya viruses circulating in the same area could have occurred during the evolution of these viruses (Figure 4). Neither apparent clustering according to the geographical area nor according to the tick species was observed, as *Ixodes scapularis*-related Bronnoya virus identified in a tick cell line falls within a clade encompassing several *Ixodes ricinus*-related BroBVs.

On a more global scale, phylogenetic analysis of the viral polymerase placed Bronnoya-related viruses, in addition to the Ubmeje virus (an unclassified *Bunyavirales* identified in Swedish *Ixodes uriae* tick in 2016-2017 [25]), in a distinct clade apparently restricted to *Ixodes* sp. ticks and positioned between the *Hantaviridae, Tospoviridae, Peribunyaviridae, Fimoviridae,* and *Cruliviridae* families. This suggests that this clade may constitute a novel family within the order *Bunyavirales* (Figure 5). An estimation of the mean evolutionary *p-distance* between recognized *Bunyavirales* families and the proposed new viral family placed this latter group in the same range of distances of families recognized by the ICTV, reinforcing the hypothesis that Bronnoya-like viruses could constitute a new family within the order *Bunyavirales* (Figure 6, Appendix A). Interestingly, more distant viruses linked to this proposed family were reported in *Dermacentor* and *Hyalomma* sp. Ticks identified in China [11] and Russia, respectively (Appendix A), suggesting a possible specialization of Bronnoya-related viruses to ticks.

### 3.2. Transmission of Bronnoya Viruses to Small Ruminants

The transmission of BroBV-like viruses by *Ixodes ricinus* ticks to the Romanian small-ruminant population upon exposure to tick bites was assessed using a LIPS assay, focusing on the predicted external domain of the BroBV glycoprotein. To determine if these sera could be considered positive and, in the absence of any positive control serum, we proposed a positivity threshold defined as the mean of signal-to-noise ratio of non-exposed French sera plus three standard deviations. No Romanian sheep or goat serum exceeded the proposed positivity threshold. However, significant differences were observed between non-exposed French sera and Romanian sheep sera collected in various villages of Tulcea County (Slava Cercheza in 2019 and 2021, and Cataloi and M. Kogalniceanu in 2021). Since these differences did not exceed the threshold, we considered these differences as artefactual and concluded that no sheep or goat serum was positive against BroBV (Figure 7).

## 4. Discussion

As part of a global study aimed at deciphering the virome of *Ixodes ricinus* ticks circulating in two Western or Eastern European countries (France and Romania), we report in this study the complete characterization of a novel group of tick-borne viruses which represents a new viral family of the order *Bunyavirales* and are primarily associated to *Ixodes* sp. ticks: Bronnoya-like viruses (BroBV-like). These viruses are negative-sense, single-stranded, bi-segmented RNA viruses lacking the small segment that encodes the viral nucleoprotein. First reported in metagenomics and meta-transcriptomics studies of the virome of *I. ricinus* ticks collected in Norway [4], Croatia [10], and Russia (Almazovo tick virus), a BroBV-related virus highly similar to those identified in *Ixodes ricinus* was identified in a *Ixodes* scapularis-derived cell line [12], questioning the exogenous nature of BroBV-associated sequences. However, Nakao et al. demonstrated the presence of an exogenous virus in the culture supernatant of an embryo-derived *Ixodes scapularis* cell line for which the size of the particles was compatible with *Bunyavirales* viruses. They also identified reads associated to BroBV in the purified supernatant. More distant viral sequences were reported in *Ixodes uriae* [25] and *Hyalomma* sp. [11] (Volzhskoe tick virus) ticks. Here, we identified six novel BroBV-like strains in *Ixodes ricinus* ticks collected on small ruminants or inenvironments with distinct biotopes.

Phylogenetic and evolutionary distance analyses placed *Ixodes*-associated Bronnoya-like viruses, including those associated to *Hyalomma* ticks, in a clade sufficiently distant from all known *Bunyavirales* families (Figure 5 and Figure 6) to constitute a novel family within the order. All members of this family seem to lack the S-segment coding for the viral nucleoprotein (NP), although one of its members is able to replicate in a tick-derived cell line [12]. The absence of this key viral protein is questionable and the origin of the NP must be further investigated, for example, by checking if another bunyavirus (or other virus)-infecting tick could complement the BroBV virus for the NP. Another hypothesis is that the S-segment is so distant to known viruses present in public databases that current homology-based tools (such as Blast or HMMER, [26]) are not able to identify the S-segment of BroBV viruses.

Within the BroBV family, the comparative phylogenetic reconstructions of the evolution history of the L and M segments showed that reassortment could have been a key event in the history of Bronnoya viruses (Figure 4). Reassortment, the phenomenon occurring during the viral replication cycle which leads to the encapsulation of viral segments of different virus origins, is a common trait of bunyaviruses [27]. It occurs when a cell (of tick or vertebrate origin) is concomitantly infected by two viruses. Reassortment plays an important role in virus evolution by generating diversity and represents a potential future source of novel viruses. In our study, we demonstrated that reassortment could have occurred between Bronnoya viruses infecting *Ixodes ricinus* (BroBV strain Romania/Tulcea 2020 tick1) and *Ixodes scapularis* (BroBV strain Cell Line ISE6) ticks, suggesting that this event could have occurred during the co-feeding of ticks on a vertebrate animal.

To go further in the characterization of the host spectrum of the new BroBV family, we tested small ruminants for the presence of antibodies directed against the glycoprotein of Romanian Bronnoya virus as indirect evidence of the ability of BroBV to infect ticks’ vertebrate hosts. To do so, of the three possible domains that we have identified using alphafold (Gn^H^, Gn^B^, and the fusion domain of Gc, Figure 2), we chose to use Gn^H^ because of its size and because the equivalent Gn^H^ domain in phenuiviruses and hantaviruses is highly immunogenic [28]. Structural studies of the GPs of other bunyaviruses show that the folding of Gn and Gc are conserved. Gn folds into two domains, called the head (GnH) and the base (GnB), while Gc is a class-II fusion protein. Although there are differences between the different families, GnH tends to be the most immunogenic [28]. The AlphaFold prediction of Bronnoya virus Gc (residues 711–1486, average pLDTT = 78) confirms that Gc is a class -II fusion protein similar to most bunyavirus fusion proteins (Figure 2). The prediction of Gn has a very low confidence with an average pLDTT of 45, but a closer inspection reveals a C-terminal region with a high pLDTT score (residues 570-664, pLDTT = 85) folded into a seven-stranded β-sandwich followed by a helical hairpin, which is the same fold observed in the GnB of other bunyaviruses [28]. Thus, although we do not have a reliable prediction of the N-terminal part (residues 1–569), by analogy to other bunyaviruses we refer to it as GnH.

No Romanian sheep or goat sera collected over a three-year surveillance period was detected as positive (Figure 7). One can argue that, without any positive control, the determination of a positive threshold is difficult. We defined the positivity threshold of the assay by reference to a likely non-exposed French control population, but we cannot be completely confident that the control population, selected in an area far from those where French BroBV was identified, was not infected by a different BroBV strain circulating in the area. Therefore, the host spectrum of Bronnoya viruses is still unknown, and further experiments are needed to determine their ability to infect multiple hosts or their restriction to tick hosts.

The spectrum of vertebrate hosts of ixodid ticks is very diverse. *Ixodes ricinus* is undoubtedly the most well-known and studied European tick species. The presence of *I. ricinus* has been recorded in more than 160 species of vertebrates [29]: larvae and nymphs usually feed on small mammals such as rodents, rabbits, birds, reptiles, bats, and lizards, whereas adults prefer larger mammals, such as sheep, cattle, dogs, deer, humans, and horses, and are the most important vector of many of European tick-borne diseases [30]. After *I. ricinus*, the second most widespread hard-tick species in Europe is *Dermacentor reticulatus* [31]. Generally, hosts of *Dermacentor* ticks include many large and small mammals, including horses, deer, cattle, dogs, cats, sheep, goats, and lagomorphs [32]. Similar to other genera of ticks, adults of *Hyalomma* sp. feed on a wide variety of mammals, mainly wild and domestic ungulates (particularly bovines), while the larvae and nymphs are specific to small mammals (leporids and insectivores) and to ground-dwelling birds of various taxonomic groups [33]. No sheep or goat sera were detected as positive for BroBV-like antibodies, but further studies are needed to explore other vertebrates, such as rodents or birds—especially the latter, which are highly abundant in the Danube Delta region [15]. In addition, a phylogenetic analysis of the polymerase protein of Bronnoya-related viruses did not allow for the identification of a link between clades and tick species. Specifically, the *Ixodes uriae*-related BroBV was more closely related to *Hyalomma*-related BroBV than to *Ixodes icinus* or *Ixodes scapularis*-related BroBV (Figure 5, Appendix A), suggesting that Bronnoya viruses seem not to be restricted to a tick species. The ability of BroBV viruses to infect multiple tick species and genera through co-feeding on the same vertebrate hosts is therefore still unknown.

## 5. Conclusions

This study is the first report of the comprehensive characterization of the evolution history and host spectrum of a new viral family within *Bunyavirales*. Members of this family represent likely tick symbionts that produce a chronic infection of the tick (since its maintenance in a tick-derived cell line was reported) and that are constitutive of the core virome specific to a given tick species and genus.

This approach, beginning with the identification of viral sequences in metatranscriptomic datasets and ending with the determination of the host spectrum of newly identified viruses, should be extended to other tick-borne viruses to obtain a better overview of the core virome of ticks and to identify viruses with arbovirus potential that could be of concern for human or animal health.

## Figures and Tables

**Figure 1 microorganisms-11-00209-f001:**
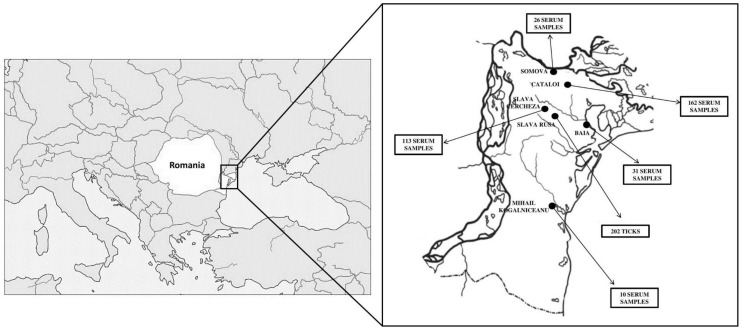
Location of small ruminants and tick specimens collected in the Danube Delta reserve.

**Figure 2 microorganisms-11-00209-f002:**
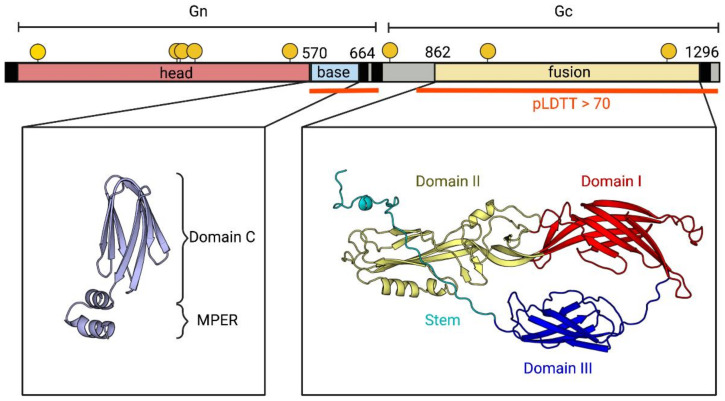
Organization of the glycoprotein. The upper panel shows a diagram of the glycoprotein precursor colored according to domains (Gn^H^ in red, Gn^B^ in cyan, and the class-II fusion protein in yellow). The boundaries of each domain are indicated on the diagram. The locations of *N*-glycosylation sites are indicated with a yellow circle on top of the diagram. The two regions that generate a reliable alphafold prediction (pLDTT > 70) are indicated with a red line below the diagram. The panels below show the alphafold predictions for these two regions colored according to domains, as indicated.

**Figure 3 microorganisms-11-00209-f003:**
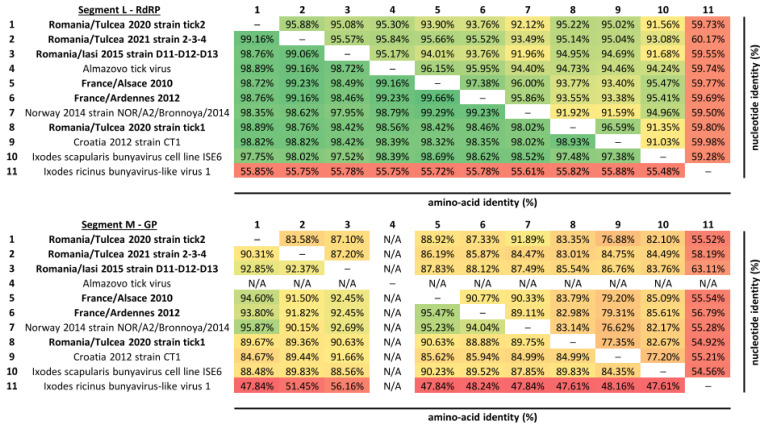
Amino-acid and nucleotide identity matrix of Bronnoya virus RNA-dependent RNA polymerase (upper panel) and glycoprotein (lower panel). Matrices were colored according to the identity scale: the lowest identities are represented in red and the highest in green. Sequences are ordered by similarity in the RdRP protein. “N/A” indicates that the glycoprotein sequence of the Almazovo tick virus is not available.

**Figure 4 microorganisms-11-00209-f004:**
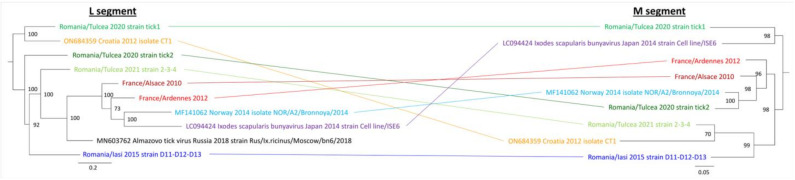
Compared phylogenetic reconstructions of Bronnoya-related viruses L (left) and M (right) segments.

**Figure 5 microorganisms-11-00209-f005:**
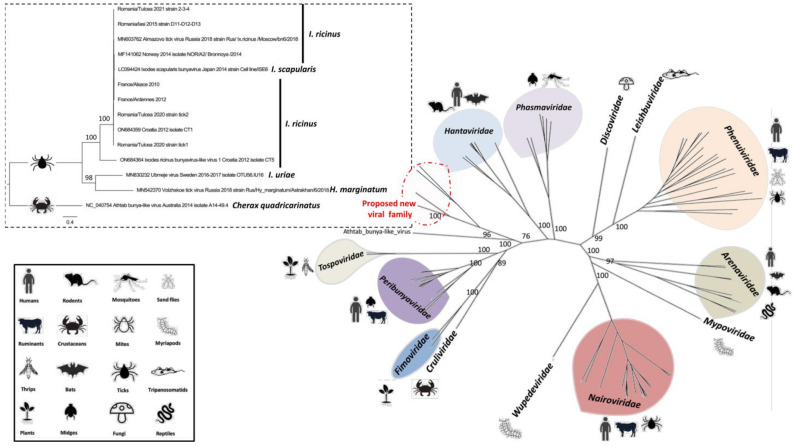
Phylogenetic reconstruction of the RNA-dependent RNA polymerase of *Bunyavirales*. Representative complete amino-acid sequences of the thirteen families comprising the order were selected (details of sequences are presented in Appendix A). Only bootstrap evaluations of the clustering of viral families within the *Bunyavirales* are shown. Inset: RdRP-based amino-acid phylogenetic relationship between members of the proposed novel virus family in relation to their tick species.

**Figure 6 microorganisms-11-00209-f006:**
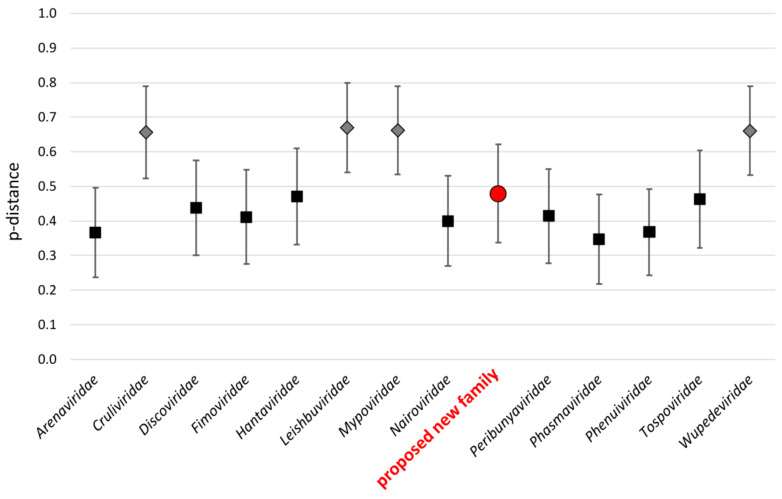
Mean evolutionary *p*-distances between *Bunyavirales* families. Families for which one sequence was used in the analysis are represented by a gray diamond, and families for which >1 sequences were included are represented by a black square. The proposed new family is highlighted by a red circle.

**Figure 7 microorganisms-11-00209-f007:**
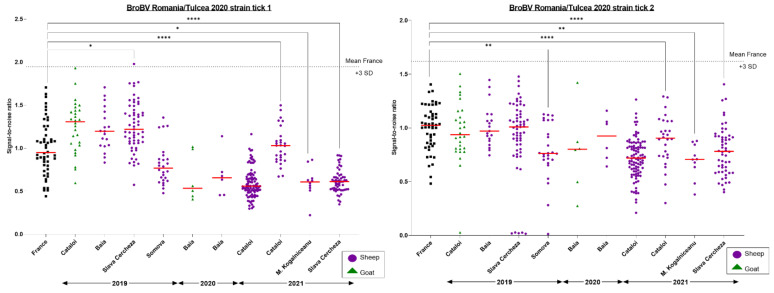
Results of the Luciferase immunoprecipitation (LIPS) small-ruminant screening against Romanian BroBV. French sheep sera (black) served as likely non-exposed negative controls. Sheep (purple) and goat (green) sera were tested for antibody responses against the BroBV Romania/Tulcea 2020 tick1 (left panel) and BroBV Romania/Tulcea 2020 tick2 strains (right panel). The ANOVA non-parametric Kruskal–Wallis test was conducted to compare each sub-population to the reference French population. Only significant differences are presented and labeled as *, **, and **** according to the level of significance of the differences.

**Table 1 microorganisms-11-00209-t001:** Ruminant serum samples collected from Tulcea County, Romania.

Year of Collection	Collection Site	Species of Small Ruminants	Number of Serum Samples
2019	Baia	Sheep	19
	Cataloi	Goats	28
		Sheep	28
	Slava Cercheză	Sheep	59
	Somova	Sheep	26
2020	Baia	Sheep	6
		Goats	6
2021	Cataloi	Sheep	106
	Mihail Kogalniceanu	Sheep	10
	Slava Cercheză	Sheep	54
	TOTAL		342

**Table 2 microorganisms-11-00209-t002:** BroBV strains identified in *I. ricinus* ticks from France and Romania.

Strain	Tick Species	Location	Year	Accession Number
Romania/Tulcea_2020_strain_tick1	*Ixodes ricinus*	Slava Rusa, Romania	2020	OQ029680 (L) OQ029681(M)
Romania/Tulcea_2020_strain_tick2			2020	OQ029682 (L) OQ029683 (M)
Romania/Tulcea_2021_strain_2-3-4			2021	OQ029684 (L) OQ029685 (M)
Romania/Iasi_2015_strain_D11-D12-D13		Iasi, Romania	2015	OQ029686 (L) OQ029687 (M)
France/Alsace_2010		Alsace, France	2010	OQ029688 (L) OQ029689 (M)
France/Ardennes_2012		Ardennes, France	2012	OQ029690 (L) OQ029691 (M)

## Data Availability

The genomes of viruses described in the study were deposited into the GenBank database under the accession numbers OQ029680, OQ029681, OQ029682, OQ029683, OQ029684, OQ029685, OQ029686, OQ029687, OQ029688, OQ029689, OQ029690, and OQ029691.

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
