# Peer review of "A Search for Tick-Associated, Bronnoya-like Virus Spillover into Sheep"

_microorganisms, 2023, doi:10.3390/microorganisms11010209_

Round 1

Reviewer 1 Report

The manuscript "A search for tick-associated Bronnoya virus spillover to sheep" deserves to be published on Microorganisms. However, it needs improvements before publication.

The introduction should be improved, as it does not contain enough elements to say that viruses related to Bunyavirales are important. However, I consider it important to mention some epidemiological or prevalence data on the ability to infect vertebrates.

- Methods can be improved. e.g., information on the determination of tick species (2.2) and expression of recombinant proteins (2.6.2) should be elaborated. Primers, strains, etc... One of my main concerns is how they detected that all the ticks collected were Ixodes ricinus, in a period from 2010-2021. I would suppose that other species have established themselves in areas where they did not exist due to climate change and global warming. Another concern is why they selected to express the head domain (GnH) to antigen design. Usually, the most immunogenic region is also the one with the greatest variability. I suggest that this should be clarified and added to the discussion.

The results and conclusions described in the manuscript are sound. However, the discussion must be improved.

Reviewer 2 Report

In this manuscript the authors describe their efforts to identify tick associated Bronnoya viruses and the potential for their spillover to sheep. While the efforts to identify any new viruses are important, there are some significant discrepancies in this manuscript. Here are the concerns:

Authors use a language throughout the manuscript, that highlights Bunyaviruses even though the focus of the manuscript is Bronnoya virus (as per the title). While Bronnoya virus is a member of Bunyavirales, what authors really report is the potential discovery of six new Bronnoya virus strains and new virus related to Bunyavirales. This should really be focus of the phrasing.

Authors’ rationalization for antigen design is not clear in the methods. The entire structure-based prediction using Alphafold didn’t really produce reliable models (average pLDTT of 45). How is the choice of Alphfold therefore of any use?

 Authors seem to use the same acronym BroBV for Bronnoya virus (Line 235) and Bronnoya-Like Viruses (Line 322). This discrepancy needs to be fixed. Due to this it becomes extremely difficult to follow the discussion.

In one paragraph of discussion authors make a case for potential reassortment events happening during co-feeding of ticks onto a vertebrate animal. They however go on to show that there is no evidence of these viruses having infected small ruminants. This needs to be reconciled with. Are the authors suggesting that there is likely another host (as per subsequent paragraphs) or are they saying this is unlikely to happen?

Round 2

Reviewer 1 Report

All my concerns were resolved and written in the manuscript.

Reviewer 2 Report

The authors have addressed the comments and reformatting of text has made the manuscript much more understandable.